# ADVERSARIALLY ROBUST TRAINING THROUGH STRUCTURED GRADIENT REGULARIZATION

## ABSTRACT

We propose a novel data-dependent structured gradient regularizer to increase the robustness of neural networks vis-a-vis adversarial perturbations. Our regularizer can be derived as a controlled approximation from first principles, leveraging the fundamental link between training with noise and regularization. It adds very little computational overhead during learning and is simple to implement generically in standard deep learning frameworks. Our experiments provide strong evidence that structured gradient regularization can act as an effective first line of defense against attacks based on long-range correlated signal corruptions.[1]

## 1 INTRODUCTION

Deep neural networks, in particular convolutional neural networks (CNNs), have been used with great success for perceptual tasks such as image classification (Simonyan & Zisserman, 2014) or speech recognition (Hinton et al., 2012). However, it has been shown that the accuracy of models obtained by standard training methods can dramatically deteriorate in the face of so-called adversarial examples (Szegedy et al., 2013; Goodfellow et al., 2014), i.e. small perturbations in the input signal, typically imperceptible to humans, that are sufficient to induce large changes in the output. This apparent vulnerability is worrisome as CNNs start to proliferate in the real-world, including in safety-critical deployments.

Although the theoretical aspects of vulnerability to adversarial perturbations are not yet fully understood, a plethora of methods has been proposed to find adversarial examples. These often transfer or generalize across different architectures and datasets (Papernot et al., 2016a; Liu et al., 2016; Tramèr et al., 2017), enabling black-box attacks even for inaccessible models. The most direct and commonly used strategy of protection is to use adversarial examples during training, see e.g. (Goodfellow et al., 2014) or (Miyato et al., 2015). This raises the question, of whether one can generalize across such examples in order to become immune to a wider range of possible adversarial perturbations. Otherwise there appears to be a danger of overfitting to a specific attack, in particular if adversarial examples are sparsely generated from the training set.

In this paper, we pursue a different route, starting from the hypothesis that the weakness revealed by adversarial examples first and foremost hints at a data scarcity problem. Inspired by the classical work of Bishop (1995) and similar in spirit to Miyato et al. (2017), we thus propose regularization as the preferred remedy to increase model robustness. Our regularization approach comes without rigid *a priori* assumptions on the model structure and filters (Bruna & Mallat, 2013) as well as strong (and often overly drastic) requirements such as layer-wise Lipschitz bounds (Cisse et al., 2017) or isotropic smoothness (Miyato et al., 2017). Instead, we will rely on the ease of generating adversarial examples to learn an *informative* regularizer, focusing on the correlation structure of the adversarial noise. Our regularizer consistently outperforms structure-agnostic baselines on long-range correlated perturbations - achieving over 25% higher classification accuracy. We thus consider it an effective first line of defense on top of which other defense mechanisms may be built.

In summary, we make the following contributions:

- We provide evidence that current adversarial attacks act by perturbing the short-range covariance structure of signals.

---

[1]Code will be made available

- We propose Structured Gradient Regularization (SGR), a data-dependent regularizer informed by the covariance structure of adversarial perturbations.

- We present an efficient SGR implementation with low memory footprint.

- We introduce a long-range correlated adversarial attack against which SGR-regularized classifiers are shown to be robust.

## 2 ADVERSARIAL ROBUST LEARNING

### 2.1 VIRTUAL EXAMPLES

Imagine we had access to a set of transformations $\tau \in \mathcal{T}$ of inputs that leave outputs invariant. Then we could augment our training data by expanding each real example $(\mathbf{x}_i, y_i)$ into *virtual examples* $(\tau(\mathbf{x}_i), y_i)$. This approach is simple and widely-applicable, e.g. for exploiting known data invariances (Schölkopf et al., 1996), and can greatly increase the statistical power of learning. A similar stochastic method has been suggested by Maaten et al. (2013): a noisy channel or perturbation $Q$ corrupts $\mathbf{x} \mapsto \tilde{\mathbf{x}}$ with probability $Q(\tilde{\mathbf{x}}|\mathbf{x})$, leading to an altered (more precisely: convolved) joint distribution $P_Q(\tilde{\mathbf{x}}, y) = P(y) \int P(\mathbf{x}|y) Q(\tilde{\mathbf{x}}|\mathbf{x}) d\mathbf{x}$.

Where could such a corruption model come from? We propose to use a generative mechanism to learn adversarial corruptions from examples. In fact, we will use a simple additive noise model in this paper,

$$Q(\tilde{\mathbf{x}}|\mathbf{x}) = Q(\boldsymbol{\xi} := \tilde{\mathbf{x}} - \mathbf{x}), \ \mathbf{E}[\boldsymbol{\xi}\boldsymbol{\xi}^\top] = \boldsymbol{\Sigma}. \tag{1}$$

Our focus will be on leveraging the correlation structure of the noise, captured by the matrix of raw second moments $\hat{\boldsymbol{\Sigma}}$, aggregated as a running exponentially weighted batch estimate computed from adversarial perturbations $\boldsymbol{\xi}$, obtained by methods such as FGSM (Goodfellow et al., 2014), PGD (Madry et al., 2017) or DeepFool (Moosavi Dezfooli et al., 2016).

### 2.2 ROBUST MULTICLASS LEARNING OBJECTIVE

Once we have an estimate of $Q$ available, we propose to optimize any standard training criterion (e.g. a likelihood function) in expectation over a mixing distribution between the uncorrupted and the corrupted data. For the sake of concreteness, let us focus on the multiclass case, where $y \in \{1, \ldots, K\}$. Define a probabilistic classifier

$$\boldsymbol{\phi} : \mathbb{R}^d \to \triangle^K, \quad \triangle^K := \{\pi \in \mathbb{R}_+^K : \mathbf{1}^\top \pi = 1\}. \tag{2}$$

For a given loss function $\ell$ such as the negative log-loss, $\ell(y, \pi) = -\log \pi_y$, our aim is to minimize

$$\mathcal{L}(\boldsymbol{\phi}) = (1 - \lambda)\mathbf{E}_{\hat{P}}\left[\ell(y, \boldsymbol{\phi}(\mathbf{x}))\right] + \lambda \mathbf{E}_{\hat{P}_Q}\left[\ell(y, \boldsymbol{\phi}(\mathbf{x}))\right], \tag{3}$$

where $\hat{P}$ denotes the empirical distribution and $\hat{P}_Q$ the noise corrupted distribution. The hyperparameter $\lambda$ gives us a degree of freedom to pay more attention to the training data vs. our (imperfect) corruption model.

### 2.3 FROM VIRTUAL EXAMPLES TO REGULARIZATION

Approaches relying on the explicit generation of adversarial examples face three *statistical challenges*: (i) How can enough diverse samples be generated to cover all attacks? (ii) How can we avoid a computational blow-up by adding too many virtual examples? (iii) How can we prevent overfitting to a specific attack? A remedy to these problems is through the use of regularization. The basic idea is simple: instead of sampling virtual examples, one tries to calculate the corresponding integrals in closed form, at least under reasonable approximations.

The general connection between regularization and training with noise-corrupted examples has been firmly established by Bishop (1995). The benefits of regularization over adversarially augmented training in terms of generalization error have been hypothesized by Galloway et al. (2018). In (Maaten et al., 2013), properties of the loss function are exploited to achieve efficient regularization techniques for certain types of noise. Here, we follow the approach pursued in (Roth et al., 2017) in using an approximation that is accurate for small noise amplitudes.

## 3 STRUCTURED GRADIENT REGULARIZATION

### 3.1 FROM CORRELATED NOISE TO STRUCTURED GRADIENT REGULARIZATION

To approximate the expectation with regard to the noisy channel, we generalize a recent approach for gradient regularization of Generative Adversarial Networks (GANs) (Roth et al., 2017), from binary classification to multiclass classification and from white noise to (adversarially) structured noise.

For the sake of simplicity of derivations we assume adversarial perturbations to be *centered*. Let us Taylor expand the log-component functions, making use of the shortcut $\psi_y := \log \phi_y$,

$$\psi_y(\cdot + \boldsymbol{\xi}) = \psi_y + \nabla \psi_y^\top \boldsymbol{\xi} + \tfrac{1}{2} \boldsymbol{\xi}^\top \triangle[\psi_y]\boldsymbol{\xi} + \mathcal{O}(\|\boldsymbol{\xi}\|^3) \tag{4}$$

For any zero-mean distribution $Q$ with second moments $\boldsymbol{\Sigma}$,

$$\mathbf{E}_Q[\psi_y(\cdot + \boldsymbol{\xi})] = \psi_y + \tfrac{1}{2}\mathrm{Tr}(\triangle[\psi_y]\boldsymbol{\Sigma}) + \mathcal{O}(\mathbf{E}[\|\boldsymbol{\xi}\|^3]). \tag{5}$$

The Hessian is calculated via the chain rule,

$$\triangle(\psi_y) = \nabla \left[ \frac{\nabla \phi_y}{\phi_y} \right] = \frac{\triangle \phi_y}{\phi_y} - \nabla \psi_y \cdot \nabla \psi_y^\top. \tag{6}$$

We will make a similar argument to (Roth et al., 2017) to show that it is reasonable to neglect the terms involving Hessians $\triangle \phi_y$. Let us therefore consider the Bayes-optimal classifier $\phi_y^*$, for which up to a normalization constant

$$\phi_y^*(\mathbf{x}) \propto P(\mathbf{x}|y)P(y), \tag{7}$$

such that

$$\mathbf{E}_P \left[ \frac{\triangle \phi_y^*(\mathbf{x})}{\phi_y^*(\mathbf{x})} \right] \propto \int \sum_y \triangle \phi_y^*(\mathbf{x}) d\mathbf{x}. \tag{8}$$

Exchanging summation and differentiation, we however have as a consequence of the normalization

$$\sum_y \triangle \phi_y(\mathbf{x}) = \triangle \left[ \sum_y \phi_y(\mathbf{x}) \right] = \triangle 1 = 0. \tag{9}$$

Thus, under the assumption that $\phi \approx \phi^*$ and of small perturbations (such that we can ignore higher order terms in Eq. (4)), we get

$$\mathbf{E}_{P_Q}[\psi_y(\mathbf{x})] = \mathbf{E}_P \left[ \psi_y(\mathbf{x}) - \tfrac{1}{2}\mathrm{Tr} \left[ \nabla \psi_y \nabla \psi_y^\top \boldsymbol{\Sigma} \right] \right] + \mathcal{O}(\mathbf{E}[\|\boldsymbol{\xi}\|^3]). \tag{10}$$

We can turn this into a regularizer by taking the leading terms (Roth et al., 2017). Identifying $P = \hat{P}$ with the empirical distribution, we arrive at the following robust learning objective:

---

**Structured Gradient Regularization (SGR)**

$$\min \boldsymbol{\phi} \to \mathcal{L}^{\mathrm{SGR}}(\boldsymbol{\phi}) = \mathbf{E}_{\hat{P}} \left[ -\log \phi_y(\mathbf{x}) \right] + \lambda \Omega_{\boldsymbol{\Sigma}}(\boldsymbol{\phi}) \tag{11}$$

$$\Omega_{\boldsymbol{\Sigma}}(\boldsymbol{\phi}) := \frac{1}{2}\mathbf{E}_{\hat{P}} \left[ \nabla_{\mathbf{x}} \log \phi_y(\mathbf{x})^\top \boldsymbol{\Sigma} \nabla_{\mathbf{x}} \log \phi_y(\mathbf{x}) \right]$$

---

For uncentered adversarial perturbations $\mathbf{E}_Q[\boldsymbol{\xi}] = \boldsymbol{\mu}$ it is easy to see that the corresponding regularizer is given by

$$\Omega_{\boldsymbol{\mu},\boldsymbol{\Sigma}}(\boldsymbol{\phi}) = -\mathbf{E}_{\hat{P}} \left[ \nabla \log \phi_y^\top(\mathbf{x}) \boldsymbol{\mu} \right] + \Omega_{\boldsymbol{\Sigma}}(\boldsymbol{\phi}). \tag{12}$$

In practice, we however observed this correction to be small.

---

**Algorithm 1** **Adversarial SGR.** Default values: $\lambda \in [0, 1]$, $\beta = 0.1$

---

**Input:** Regularization strength $\lambda$, exponential decay rate $\beta$, batch size $m$
Initial classifier parameters (weights and biases) $\omega_0$
   **while** $\omega_t$ not converged **do**
      Sample minibatch of data $\{(\mathbf{x}^{(1)}, y^{(1)}), ..., (\mathbf{x}^{(m)}, y^{(m)})\} \sim \hat{\mathbb{P}}$.
      Compute (normalized) adversarial perturbations $\boldsymbol{\xi}^{(1)}, ..., \boldsymbol{\xi}^{(m)}$
      Compute covariance matrix or covariance function $\mathrm{Cov}(\boldsymbol{\xi}^{(1)}, ..., \boldsymbol{\xi}^{(m)})$
      Update running average $\hat{\boldsymbol{\Sigma}}_t \leftarrow (1-\beta)\,\hat{\boldsymbol{\Sigma}}_{t-1} + \beta\,\mathrm{Cov}(\boldsymbol{\xi}^{(1)}, ..., \boldsymbol{\xi}^{(m)})$

$$\mathcal{L}(\omega) = \frac{1}{m} \sum_{i=1}^{m} \left[ \sum_{k=1}^{K} -y_k^{(i)} \log \phi_k(\mathbf{x}^{(i)}; \omega) \right]$$

$$\Omega_{\hat{\boldsymbol{\Sigma}}_t}(\omega) = \frac{1}{2m} \sum_{i=1}^{m} \left[ \sum_{k=1}^{K} y_k^{(i)} \nabla_{\mathbf{x}} \log \phi_k(\mathbf{x}^{(i)}; \omega)^{\top} \hat{\boldsymbol{\Sigma}}_t \nabla_{\mathbf{x}} \log \phi_k(\mathbf{x}^{(i)}; \omega) \right]$$

$$\omega_t \leftarrow \omega_{t-1} - \nabla_\omega \Big( \mathcal{L}(\omega) + \lambda\, \Omega_{\hat{\boldsymbol{\Sigma}}_t}(\omega) \Big) \Big|_{\omega_{t-1}}$$

   **return** $\omega_t$

---

The gradient-descent step can be performed with any learning rule. We used Adam in our experiments. See Sec. 3.3 for an efficient implementation of the regularizer.

## 3.2 PROPERTIES

There are a few facts that are important to point out about the derived SGR regularizer.

(i) As the regularizer provides an efficiently computable approximation for an intractable expectation, it is clearly *data-dependent*. $\Omega_{\boldsymbol{\Sigma}}$ penalizes loss-gradients evaluated at training points. This is different from standard regularizers that penalize some norm of the parameter vector, such as $L_2$-regularization. In the latter case, we would expect the regularizer to have the largest effect in the empty parts of the input space, where it should reduce the variability of the classifier outputs. On the contrary, $\Omega_{\boldsymbol{\Sigma}}$ has its main effect around the data manifold. In this sense, it is complementary to parameter regularization.

(ii) The SGR regularizer is *intrinsic* in the sense that it does not depend on the parameters of the classifier (for a good reason, we have not mentioned any parameterization), but instead directly acts on the function realized by the classifier $\phi$ (the relevant gradients measure the sensitivity of $\phi$ with regard to the input $\mathbf{x}$ and not with regard to parameters). Thus, it is parameterization invariant and can be naturally applied to any function space, whether it is finite-dimensional or not.

(iii) We can gain complementary insights into SGR by explicitly computing it in terms of classifier logits $\varphi_y(\mathbf{x})$. As outlined in Sec 7.1 in the Appendix, we obtain the following expression for the structured gradient regularizer

$$\Omega_{\boldsymbol{\Sigma}}(\phi) = \frac{1}{2} \mathbf{E}_{\hat{P}} \Big[ (\nabla \varphi_y - \langle \nabla \varphi \rangle)^{\top} \boldsymbol{\Sigma} (\nabla \varphi_y - \langle \nabla \varphi \rangle) \Big] \ , \langle \nabla \varphi \rangle\,(\mathbf{x}) := \sum_y \nabla \varphi_y(\mathbf{x}) \phi_y(\mathbf{x}) \qquad (13)$$

We can therefore see that SGR is penalizing large *variations* of the class-conditional logit-gradients $\nabla \varphi_y$ around their data-dependent average $\langle \nabla \varphi \rangle$. For simple one-layer softmax classifiers we obtain $\Omega_{\boldsymbol{\Sigma}}(\phi) = 1/2\, \mathbf{E}_{\hat{P}} \big[ (\omega_y - \langle \omega \rangle)^{\top} \boldsymbol{\Sigma}\, (\omega_y - \langle \omega \rangle) \big]$. This suggests an intriguing connection to variance-based regularization. Weight-decay regularization, on the other hand, simply penalizes large norms.

## 3.3 IMPLEMENTATION

The implementation of SGR training is outlined in Algorithm 1. The matrix of raw second moments is aggregated through a running exponentially weighted average

$$\hat{\boldsymbol{\Sigma}}_t \leftarrow (1-\beta)\,\hat{\boldsymbol{\Sigma}}_{t-1} + \beta\,\mathrm{Cov}(\boldsymbol{\xi}^{(1)}, ..., \boldsymbol{\xi}^{(m)}) \qquad (14)$$

with the decay rate $\beta$ as a tuneable parameter trading off weighting between current ($\beta \to 1$) and past ($\beta \to 0$) batch averages. For low-dimensional data sets, we can directly compute the full matrix

$$\mathrm{Cov}(\boldsymbol{\xi}^{(1)}, ..., \boldsymbol{\xi}^{(m)}) = \frac{1}{m} \sum_{i=1}^{m} \boldsymbol{\xi}_i \boldsymbol{\xi}_i^{\top} \, . \tag{15}$$

For high-dimensional data sets, we have to resort to more memory-efficient representations leveraging the sparseness of the covariance matrix and covariance-gradient matrix-vector product. For image data sets, the covariance matrix can be estimated as a function of the displacement between pixels, as illustrated in Fig. 2,

$$\left[ \mathrm{Cov}(\boldsymbol{\xi}^{(1)}, ..., \boldsymbol{\xi}^{(m)}) \right]_{ij} \simeq \mathrm{CovFun}\left( ||\mathbf{i} - \mathbf{j}||_2 \right) \tag{16}$$

where $\mathbf{i} = (i_r, i_c)$ denotes the 2D pixel location and $i$ is the covariance matrix index obtained by flattening the 2D pixel location into a 1D array such that $i_r$ and $i_c$ denote the row and column numbers of pixel $\mathbf{i}$, and similar for $\mathbf{j}$ respectively. Rounding pixel displacements $||\mathbf{i} - \mathbf{j}||_2$ to the nearest integer, the covariance function simply becomes an array of real numbers storing the current estimate of the covariance between two pixels displaced by the difference in array indices.

In our experiments, we normalize the perturbations such that the average diagonal entry of the covariance matrix is one, thus allowing for more intuitive optimization over the regularization strength parameter $\lambda$. We note also that it is not necessary to update $\hat{\boldsymbol{\Sigma}}$ at every training step and it is even possible to train with a fixed "expert"-designed covariance matrix.

The other crucial quantities needed to evaluate the regularizer are $\log \phi_k(\cdot)$ for training points $(\mathbf{x}^{(i)}, y^{(i)})$. This is simply the per-sample cross-entropy loss, which is often available as a highly optimized callable operation in modern deep learning frameworks, reducing the implementation of the SGR regularizer to merely a few lines of code at very little computational overhead.

# 4 EXPERIMENTS

## 4.1 EXPERIMENTAL SETUP

**Classifier architectures & data preprocessing.** We trained Convolutional Neural Networks (CNNs) with seven hidden layers (and different numbers of filters for each data set), on CIFAR10 (Krizhevsky & Hinton, 2009) and MNIST (LeCun, 1998). Our models are identical to those used in (Carlini & Wagner, 2017; Papernot et al., 2016b). For both data sets, we adopt the following standard preprocessing and data augmentation scheme (He et al., 2016): each training image is zero-padded with four pixels on each side[2], randomly cropped to produce a new image with the original dimensions and horizontally flipped with probability one half[3]. We also standardize each image to have zero mean and unit variance when passing it to the classifier. Further details can be found in the Appendix.

**Training methods.** We train each classifier with a number of different training methods: (i) clean, i.e. through the standard softmax cross-entropy objective, (ii) clean $+ L_2$ weight decay regularization, (iii) adversarially augmented training, i.e. training the classifier on a mixture of clean and adversarial examples (iv) GN (gradient-norm) regularized (corresponding to SGR with identity covariance matrix) and (v) SGR regularized. The SGR covariance matrix was computed from either: (a) loss gradients, (b) sign of loss gradients (corresponding to FGSM) or (c) PGD perturbations. The performance of these SGR variants was identical in all our experiments, hence we collectively report them as SGR.

**Adversarial attacks.** We evaluate the classification accuracy against several adversarial attacks: Fast Gradient Sign Method (FGSM) (Goodfellow et al., 2014), Projected Gradient Descent (PGD) (Madry et al., 2017) and DeepFool (Moosavi Dezfooli et al., 2016). The attack strength $\epsilon$ is reported in units of $1/255$. The SNR reported for the DeepFool attack is computed according to Eq. (2) in (Moosavi Dezfooli et al., 2016). The attacks were implemented with the open source CleverHans Library (Papernot et al., 2017). Further details and attack hyper-parameters can be found in Sec. 7.2 in the Appendix.

---

[2]We shifted the CIFAR10 data by $-0.5$ so that padding adds mid rgb-range pixels.

[3]We do not flip MNIST digits.

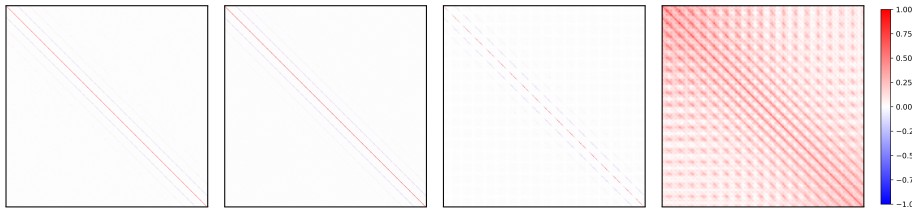

Figure 1: Covariance matrices of PGD, FGSM and DeepFool perturbations as well as CIFAR10 training set (for comparison). The short-range structure of the perturbations is clearly visible. It is also apparent that the first two attack methods yield perturbations with almost identical covariance structure. We are showing a center-crop for better visibility (25% trimmed on each side). For comparison the matrices were rescaled such that their maximum element has an absolute value of one.

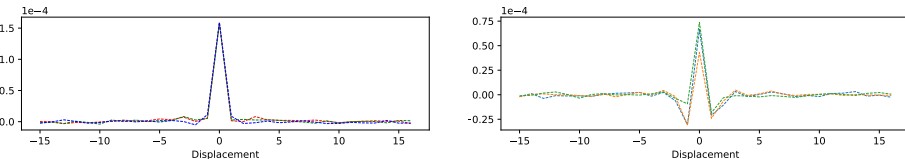

Figure 2: PGD covariance functions: (Left) intra-channel covariance functions, measuring correlations between identical color channels, (the coloring in the plot matches the corresponding channel), (right) inter-channel covariance functions, measuring correlations between distinct color channels. The rapid decay with pixel displacement, indicating very short decay length, is clearly visible.

## 4.2 Covariance Structure of Adversarial Perturbations

To investigate the covariance structure of adversarial perturbations, we trained a clean classifier for 50 epochs and computed adversarial perturbations for every data point in the test set. The perturbations were flattened into 1D arrays before computing the full covariance matrix[4] as in Eq. (15).

The short-range correlation structure of the perturbations, shown in Fig. 1, is clearly visible: the correlations decay much faster than those of the data set covariance matrix. Figure 2 shows the corresponding PGD covariance functions. The rapid decay with pixel displacement, indicating very short decay length[5], is again evident. It thus seems that an unregularized classifier vulnerable to adversarial perturbations gives too much weight to short-range correlations (low-level patterns) and not enough weight to long-range ones (globally relevant high-level features of the data).

## 4.3 Long-range Correlated Noise Attack

To investigate the effect and potential benefit of using a structured covariance matrix in the SGR regularizer versus an "unstructured" diagonal covariance, corresponding to gradient-norm regularization, we compute the classification accuracy against a worst-of-100 attack in which the adversary samples perturbations from a long-range correlated multivariate Gaussian with covariance matrix specified through an exponentially decaying covariance function parametrized by a variable decay length $\zeta$. Inspired by the PGD covariance function, we chose intra-channel CovFun $f(r) = \exp(-r/\zeta)$ and inter-channel CovFun $f(r) = 0.5 \exp(-r/\zeta)$. The corresponding covariance matrices are depicted in Fig. 5 in the Appendix. LRC samples are shown in Fig. 3.

The worst-of-100 attack then consists in perturbing every test set data point 100 times and evaluating the classifier accuracy against the worst of these perturbations. We tested for several decay lengths in the range $\zeta \in [1, 2, 4, 8, 16]$ and attack strengths $\epsilon \in [0.01, 0.7]$ with which the perturbations were scaled. Figure 4 in the Appendix shows the accuracy of different models as a function of $\epsilon$. As a baseline, we also trained a model on LRC-augmented input. For SGR and GN we performed a hyper-parameter search over $\lambda$ (at fixed $\zeta = 8$) and report results for the best performing model.

---

[4]Strictly speaking, $\hat{\Sigma}$ denotes the matrix of raw second moments. We will refer to $\hat{\Sigma}$ as the covariance matrix however, since the difference between the two is negligible.

[5]Decay length is defined as the displacement over which the covariance function decays to $1/e$ of its value.

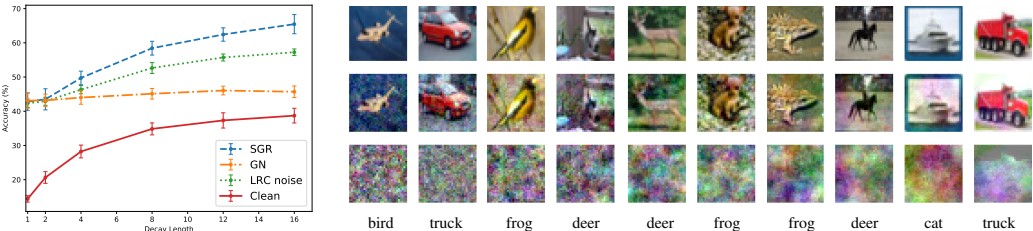

Figure 3: (Left) Accuracy of different models as a function of the LRC attack decay length $\zeta \in [1, 16]$ at fixed $\epsilon = 0.3$ (averaged over five runs). The plot clearly demonstrates the superiority of using a structured covariance matrix in the SGR regularizer. (Right) LRC perturbed samples with increasing decay lengths $\zeta \in [1, 2, 4, 8, 12]$ (two examples per decay length). The top row shows original test set images, the middle row shows adversarially perturbed samples, while the bottom row shows the adversarial perturbations rescaled to fit the full range of rgb values (their actual range and hence saliency is smaller). The footer indicates the classifier predictions on adversarial input.

The LRC attack is a natural prototype for *low frequency perturbations*, as opposed to existing attacks which we have shown to mainly corrupt the short range (high frequency) structure of signals. As shown in Fig. 3 (left), the low frequency range is precisely the regime where the benefit of structured regularization comes into play: SGR clearly and consistently outperforms GN on long-range correlated perturbations, achieving over $25\%$ higher classification accuracy on LRC-perturbed input with $\zeta \sim 16$. As the decay length goes to zero, the synthetic covariance matrix converges to the identity matrix and SGR performance approaches GN performance.

Finally, to verify that those effects are not a result of our synthetically chosen covariance function, we also performed the same experiments with the covariance matrix computed from the CIFAR10 training set, shown in Fig. 1. The accuracies for $\epsilon = 0.3$ are: SGR $65.8\%$, GN $48.7\%$, LRC noise $57.9\%$ and clean $46.5\%$, again confirming the benefit of using a structured covariance in the regularizer.

### 4.4 WHITE-BOX & TRANSFER ATTACK ACCURACY

A summary of the white-box and transfer attack accuracies can be found in Table 1. We train all models to achieve the highest possible white-box / transfer attack accuracies, averaged over $\epsilon$ in the range $[0, 32]$ for the MNIST and $[0, 8]$ for the CIFAR10 test set, i.e. the best performing models were selected according to the integrated area under the attack curve. The corresponding hyper-parameters are listed in Sec. 7.2 in the Appendix.

On MNIST, SGR/GN achieve PGD white-box accuracies of $96.5\%$ compared to the $78.3\%$ accuracy achieved by the clean model. On CIFAR10, SGR/GN trained models achieve an accuracy of $53.4\%$ against targeted T-PGD attack (target is selected uniformly at random), whereas the clean model achieves $24.9\%$ and PGD-trained model achieves $69.7\%$ accuracy. SGR/GN trained models thus achieve white-box accuracies that are intermediate between those of the clean and adversarially trained models. The higher white-box accuracies achieved by adversarially trained models, however comes at the expense of a drop in test set accuracies. On CIFAR10, the test set accuracy of the PDG-trained model drops to $77.7\%$, while SGR/GN achieve $83.4\%$ and $83.7\%$ respectively.

An interesting observation can be made from the transfer attack accuracies (bold-face entries) on CIFAR10. As can be seen, the PGD adversarial perturbations obtained from SGR/GN trained models hurt the clean model more ($60.3\%$) than the PGD perturbations obtained from PGD/FGSM adversarially trained models ($77.0\%/81.9\%$). Moreover, SGR achieves significantly higher accuracies against PGD transfer attacks derived from the clean model ($80.6\%$) than the clean model does on PGD perturbations obtained from the SGR-trained model ($60.3\%$), while accuracies achieved by SGR models on PDG samples from PGD-trained models ($75.8\%$) are comparable to those in the other direction ($74.0\%$).

As expected, the difference between SGR and GN in terms of white-box and transfer attack accuracies is less significant in this experiment, which can be explained by the evident short-range nature of current adversarial attacks (see discussion in Sec. 4.2). Reassuringly, we did not observe any loss of

MNIST $\epsilon = 32$

| TRAIN. M. | TEST | RAND | PGD | T-PGD | CLEAN | PGD | FGSM | GN | SGR | FOOL |
|---|---|---|---|---|---|---|---|---|---|---|
| CLEAN | 99.4 | 99.3 | 78.3 | 74.1 | **98.3** | **98.9** | **98.9** | 97.9 | 98.3 | 0.144 |
| WDECAY | 98.8 | 98.7 | 92.5 | 95.0 | **98.4** | **98.4** | **98.4** | 96.9 | 97.6 | 0.229 |
| PGD | 99.5 | 99.5 | 99.1 | 99.3 | **99.4** | 99.3 | **99.4** | 99.3 | 99.3 | 0.453 |
| FGSM | 99.4 | 99.4 | 98.5 | 98.0 | **99.3** | **99.2** | **99.2** | 99.2 | 99.2 | 0.339 |
| GN | 99.5 | 99.5 | 96.5 | 97.6 | **99.3** | **99.2** | **99.2** | 98.0 | 98.4 | 0.309 |
| SGR | 99.6 | 99.5 | 96.6 | 96.5 | **99.4** | **99.3** | **99.4** | 98.5 | 98.7 | 0.303 |

CIFAR10 $\epsilon = 8$

| TRAIN. M. | TEST | RAND | PGD | T-PGD | CLEAN | PGD | FGSM | GN | SGR | FOOL |
|---|---|---|---|---|---|---|---|---|---|---|
| CLEAN | 84.7 | 83.2 | 28.8 | 24.9 | **72.2** | **77.0** | **81.9** | 61.9 | 60.3 | 0.010 |
| WDECAY | 81.4 | 79.2 | 27.1 | 32.3 | **69.7** | **71.7** | **68.9** | 54.1 | 52.9 | 0.015 |
| PGD | 77.7 | 77.7 | 62.4 | 69.7 | **77.1** | 71.5 | **76.2** | 74.0 | 74.0 | 0.061 |
| FGSM | 81.9 | 82.0 | 55.2 | 66.4 | **81.1** | 75.0 | **78.0** | 76.5 | 76.4 | 0.046 |
| GN | 83.7 | 83.7 | 41.9 | 54.1 | **81.6** | **76.9** | **82.0** | 68.1 | 67.9 | 0.039 |
| SGR | 83.4 | 82.8 | 41.5 | 53.4 | **80.6** | **75.8** | **81.2** | 66.8 | 65.5 | 0.039 |

Table 1: Test set, white-box and PGD-transfer attack accuracies. Each row corresponds to a model trained with a different training method, evaluated against the respective white-box (non-bold entries) and PGD-transfer attacks (bold-face entries) indicated above each column (T-PGD refers to targeted attack). The white-box and transfer attack accuracies are averaged over attack strengths in the range $\epsilon \in [0, 32]$ for MNIST and $\epsilon \in [0, 8]$ for CIFAR10, i.e. the reported accuracies represent the integrated area under the attack curve. For the transfer attack accuracies of a training method against itself, we trained a separate model with the same hyper-parameters to obtain the perturbations.

performance in all our experiments when using the covariance function instead of the full covariance matrix in the SGR regularizer.

## 5 RELATED WORK

A large body of related work on distributionally robust optimization (DRO) (Sinha et al., 2017; Gao & Kleywegt, 2016) seeks a classifier that minimizes the worst-case loss against an adversary that can shift the entire training set within an uncertainty ball around the empirical distribution. Adversarial training can be viewed as a special case of DRO against a pointwise adversary that independently perturbs each example. While these methods aim at identifying the worst-case distribution, we propose to efficiently approximate expectations over corrupted distributions through structure-informed regularization.

Coincidentally, Ross & Doshi-Velez (2017) and Simon-Gabriel et al. (2018) also suggested to use gradient-norm regularization to improve the robustness against adversarial attacks. While (Simon-Gabriel et al., 2018) investigated the effect of different gradient-norms induced by constraints on the magnitude of the perturbations, we focused on a *structured* generalization of gradient regularization.

Similar in spirit to our work is also the recent work on adversarial training vs. weight decay regularization (Galloway et al., 2018). While weight decay acts on the parameters, our regularizer acts on the function realized by the classifier. As discussed in Sec. 3.2, the two approaches are complementary to each other and can also be combined.

From a Bayesian perspective, regularization can be understood as imposing a prior distribution over the model parameters. As parameters in neural networks are non-identifiable, we ultimately prefer to impose priors in the space of functions directly (Teh, 2017; Neal, 2012; 1996). As pointed out in Sec. 3.2, this is indeed one of the key properties of our regularizer.

## 6 CONCLUSION

The fact that adversarial perturbations can fool classifiers while being imperceptible to the human eye, hints at a fundamental mismatch between how humans and classifiers try to make sense of the data they see. We provided evidence that current adversarial attacks act by perturbing the short-range correlations of signals. We proposed a novel *structured gradient regularizer* (SGR) informed by the covariance structure of adversarial noise and presented an efficient SGR implementation with low memory footprint. Devising further (e.g. gradient-based) low frequency attacks is an interesting direction of future research.

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

## 7 APPENDIX

### 7.1 SGR PROPERTIES

(iii) We can gain complementary insights into SGR by explicitly computing it in terms of logits

$$\phi_y(\mathbf{x}) = \frac{\exp\left(\varphi_y(\mathbf{x})\right)}{\sum_y \exp\left(\varphi_y(\mathbf{x})\right)} \tag{17}$$

Defining the class average of the logit-gradients as

$$\langle \nabla \varphi \rangle (\mathbf{x}) := \sum_y \nabla \varphi_y(\mathbf{x}) \phi_y(\mathbf{x}) \tag{18}$$

we obtain the following expression for the structured gradient regularizer

$$\Omega_{\boldsymbol{\Sigma}}(\phi) = \frac{1}{2} \mathbf{E}_{\hat{P}} \left[ (\nabla \varphi_y - \langle \nabla \varphi \rangle)^\top \boldsymbol{\Sigma} (\nabla \varphi_y - \langle \nabla \varphi \rangle) \right] \tag{19}$$

We can therefore see that SGR is penalizing large *fluctuations* of the class-conditional logit-gradients $\nabla \varphi_y$ around their data-dependent class average $\langle \nabla \varphi \rangle$. For simple one-layer softmax classifiers $\varphi_y(\mathbf{x}) = \omega_y^\top \mathbf{x} + b_y$, we obtain

$$\Omega_{\boldsymbol{\Sigma}}(\phi) = \frac{1}{2} \mathbf{E}_{\hat{P}} \left[ (\omega_y - \langle \omega \rangle)^\top \boldsymbol{\Sigma} (\omega_y - \langle \omega \rangle) \right] \tag{20}$$

This suggests an intriguing connection to variance-based regularization. Weight-decay regularization, on the other hand, simply penalizes large norms.

### 7.2 HYPER-PARAMETERS

| Attack | Hyper-parameters |
|--------|------------------|
| FGSM | $\epsilon$ : in units of $1/255$ |
| PGD | $\epsilon$ : in units of $1/255$
$\epsilon_{\text{iter}} = \epsilon/5$
nb_iter = 10 (wb) , 40 (t) |
| DEEP FOOL | overshoot : 0.02
max_iter= 100 |

Table 2: Attack hyper-parameters. wb stands for white-box, t for transfer attacks.

An overview of the attack hyper-parameters can be found in Table 2. Untargeted attacks find adversarial perturbations to an arbitrary one of the $K-1$ other classes. Both PGD and DeepFool are iterative attacks: they use 10 / 40 (for the white-box / transfer attack tables) and 100 iterations respectively. The numbers reported for the DeepFool attack are computed according to Eq. (2) in (Moosavi Dezfooli et al., 2016). The adversarial example construction processes use the most likely label predicted by the classifier in order to avoid label leaking (Kurakin et al., 2016) during adversarially augmented training. The perturbed images returned by the attacks were clipped to lie within the original rgb range, as is commonly done in practice. The attacks were implemented with the open source CleverHans Library (Papernot et al., 2017).

For SGR/GN we performed a hyper-parameter search over the regularization strength $\lambda$ and SGR decay parameter $\beta$. For adversarial training, we trained models with $\epsilon$ in the full range $\epsilon \in [0, 32]$ for MNIST and $\epsilon \in [0, 8]$ for CIFAR10 and report results for the best performing one. The hyper-parameters of the best performing models are: CIFAR10: SGR $\lambda = 0.005$, $\beta = 0.1$, GN $\gamma = 0.025$, PGD $\epsilon = 8$, FGSM $\epsilon = 4$, weight-decay 0.005. MNIST: SGR $\lambda = 3.0$, $\beta = 0.1$, GN $\gamma = 10.0$, PGD $\epsilon = 32$, FGSM $\epsilon = 32$, weight-decay 0.005.

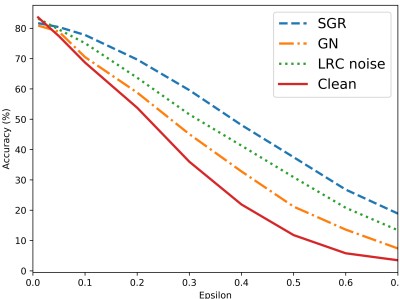

Figure 4: Long Range Correlated Noise. Classification accuracy of different models as a function of the attack strength $\epsilon$ for an LRC attack with a fixed decay length $\zeta = 8$ on CIFAR10. The SGR/GN hyper-parameters are reported in Sec. 4.3. See Fig. 3 for a plot of the accuracy of different models as a function of the LRC attack decay length $\zeta \in [1, 16]$ at fixed $\epsilon = 0.3$

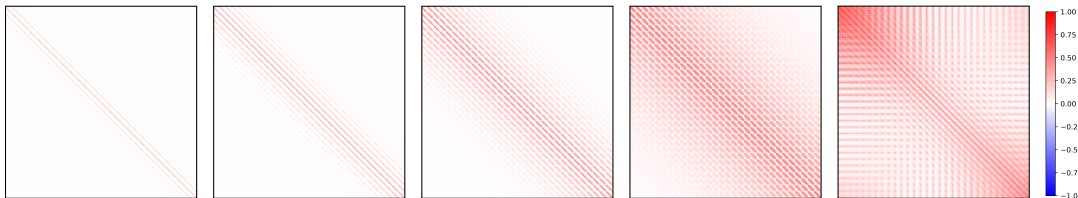

Figure 5: Long-range structured covariance matrices corresponding to the covariance functions defined in Sec. 4.3 for increasing decay lengths $\zeta \in [1, 2, 4, 8]$, as well as covariance matrix of CIFAR10 training set (for comparison). As an alternative to the covariance function representation described in Sec. 3.3, we can also leverage the block-structure of the full covariance matrix to obtain a sparse representation: as can be seen in the figures, different equally distant pairs of rows are often nearly identically correlated. Hence it is enough to estimate the covariance between a few pairs of rows $r_m$ and $r_n > r_m$ (until the correlations become sufficiently small). This amounts to for instance estimating the covariance matrix by a diagonal and a few off-diagonal blocks.

## 7.3 FURTHER EXPERIMENTAL RESULTS

**Smoothness of Regularized Classifier**   At the most basic level of abstraction, adversarial vulnerability is related to an overly-steep classifier gradient, such that a tiny change in the input can lead to a large perturbation of the classifier output. In order to analyze this effect, we visualize the softmax activations along linear trajectories between various clean and corresponding adversarial examples. As can be seen in Fig. 6, the regularized classifier consistently leads to smooth classifier outputs.

**Label leaking.**   Label leaking can cause models trained with adversarial examples to perform better on perturbed than on clean inputs, as the model can learn to exploit regularities in the adversarial example construction process (Kurakin et al., 2016). This hints at a danger of overfitting to the specific attack when training with adversarial examples. SGR regularized classifiers, on the other hand, do not suffer from label leaking, as the adversarial examples are never directly fed to the classifier.

## 7.4 CLASSIFIER ARCHITECTURES

Networks were trained for $50/75$ epochs on MNIST/CIFAR10 using the Adam optimizer (Kingma & Ba, 2014) with learning rate $0.001$ and minibatch size $128$.

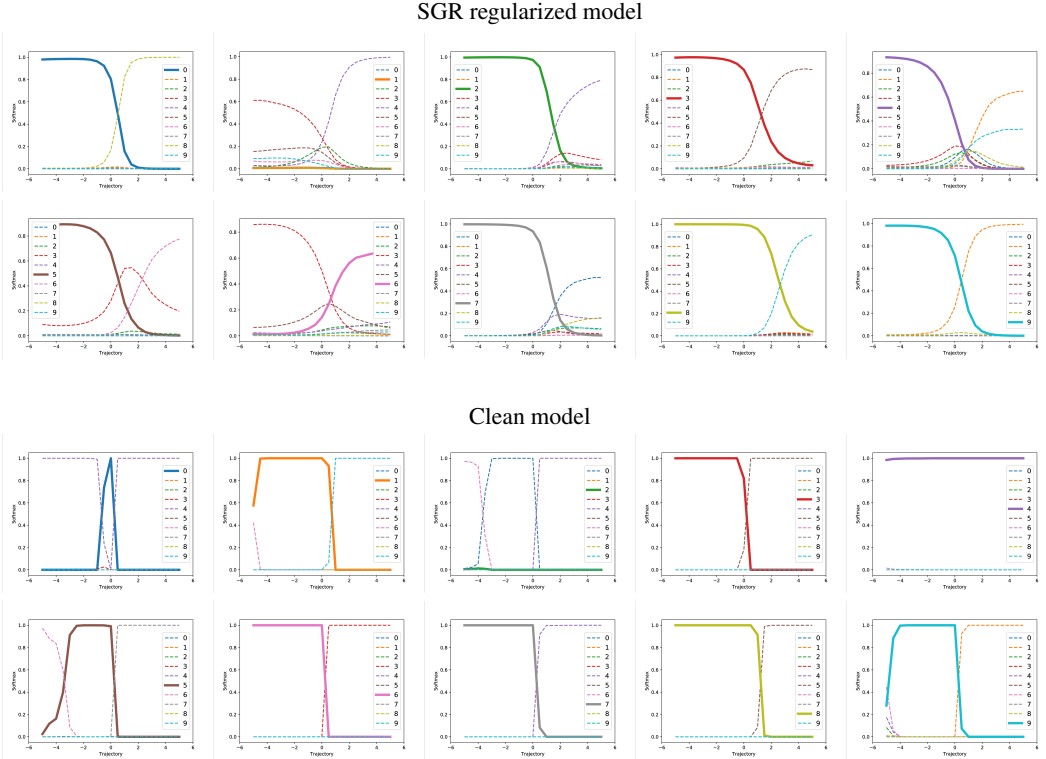

Figure 6: Softmax activations along a linear trajectory between a clean sample and plus/minus five times the $\epsilon = 4$ PGD adversarial perturbation for an SGR regularized classifier (top) and a clean classifier (bottom) on CIFAR10.

| Classifier | |
|---|---|
| Feature Block | Conv2D (filter size: $3 \times 3$, feature maps: params[0], stride: $1 \times 1$) 
 ReLU |
| Feature Block | Conv2D (filter size: $3 \times 3$, feature maps: params[1], stride: $1 \times 1$) 
 ReLU |
| MaxPooling | MaxPool (pool size: (2,2)) |
| Feature Block | Conv2D (filter size: $3 \times 3$, feature maps: params[2], stride: $1 \times 1$) 
 ReLU |
| Feature Block | Conv2D (filter size: $3 \times 3$, feature maps: params[3], stride: $1 \times 1$) 
 ReLU |
| MaxPooling | MaxPool (pool size: (2,2)) |
| Fully-Connected | Dense (units: params[4]) |
| Fully-Connected | Dense (units: params[5]) |
| Fully-Connected | Dense (units: params[6]) 
 Softmax |

Table 3: Classifier Architectures: CIFAR10 params=$[64, 64, 128, 128, 256, 256, 10]$, MNIST params= $[32, 32, 64, 64, 200, 200, 10]$. Our models are identical to those used in (Carlini & Wagner, 2017; Papernot et al., 2016b).

