# OpenReview forum: "Adversarially Robust Training through Structured Gradient Regularization"
_ICLR.cc/2019/Conference_

### Official Review · AnonReviewer1 · 2018-11-01
**simple and reasonable idea, somewhat unconvincing theoretical analysis, weak experiments**

**Rating:** 3
**Confidence:** 4

**Review:**

Summary of the paper:
This paper proposes to use structured gradient regularization to increase adversarial robustness of neural network. Here, the gradient regularization is to regularize some norm of the gradients on neural network input. "structured" means that instead of just minimizing the L2 norm of the gradients, a "mahalanobis norm" is minimized. The covariance matrix is updated continuously to track the "structure" of gradients/perturbations. Whitebox attack and blackbox attack

The paper is well written, both theory and experiments are well explained. The analysis of LRC attack on SGR trained models are interesting.

However, I believe the paper has major flaws in several aspects.

The whitebox robustness evaluation is weak. Whitebox PGD with 10 iterations is not enough for discovering true robustness of a neural network, which makes the experiments unconvincing. PGD with 100 iterations and 50 random starts would make the evaluation much convincing wrt to whitebox attack. https://github.com/MadryLab/mnist_challenge
I noticed that in Table 1, the authors reported averaged results across different epsilons. Although I see the motivation to give equal weights to small and large perturbations, it makes it hard to compare with previous papers. I think the authors should a least report commonly used eps in the literature, including MNIST eps=0.1, 0.2, 0.3 and CIFAR10 eps=8/255. Currently, for MNIST eps=32/255=0.125 is much below the standard eps for benchmarking MNIST.

In my opinion, when evaluating robust optimization / gradient regularization methods, robustness under the strongest whitebox should be the major benchmark. Because "intrinsic" robustness is their goal. In contrast, black-box results are less important. This is because 1) evaluating black-box robustness on a few attacks hardly give any conclusive statements; 2) if we're pursuing black-box robustness, there're many randomization methods that boosts black-box robustness under various settings. How does a gradient regularization method help on top of those should be at least evaluated.
So if the paper wants to claim black-box robustness, it needs at least include experiments like 2), so it provides useful benchmarks to practitioners.

There're also a few problems in the motivation / analysis.
"""A remedy to these problems is through the use of regularization. The basic idea is simple: instead of sampling virtual examples, one tries to calculate the corresponding integrals in closed form, at least under reasonable approximations."""
The adversarial robustness problem is not about integral over a neighborhood, it is about the maximum loss over a neighborhood. This is likely why previous attempts on gradient regularization and adversarial training on FGSM attack fails. And the success is of PGD training is largely due to that the loss minimize over the adversarial example that gives the maximum loss.

"""Thus, under the assumption that \phi \approx \phi^* and of small perturbations (such that we can ignore higher order terms."""
The Bayes optimal assumption seems to be arbitrary to me. If \phi is nearly Bayes-optimal, why would we worry about adversarial examples?



Other relatively minor problems

In the caption of Figure 1, """Covariance matrices of PGD, FGSM and DeepFool perturbations as well as CIFAR10 training set (for comparison). The short-range structure of the perturbations is clearly visible. It is also apparent that the first two attack methods yield perturbations with almost identical covariance structure."""
PGD and FGSM have very different attack power. If they are similar by any measure, wouldn't that mean the measure (covariance structure) is too coarse?

In Section 3.1, the paper talks about both centered and uncentered adversarial examples.
I assumed that the authors mean that the distribution of perturbations are centered?
First, I think this the authors should make this more explicit.
Second, I think this is not a realistic to assume the perturbations to be centered, because for image data, the epsilon-ball usually intersects with data domain boundary. So I'm wondering in the experiments, which version was used? centered or uncentered?

Figure 5 shows periodic patterns on covariance matrices. I didn't find explanation of the periodic patterns in the covariance matrices. It would nice if the authors can explain it or point me the relevant sections in the paper.

I don't fully get the idea of LRC attack. Is it purely sampling? are there optimization involved?

Figure 3, I suggest the authors show perturbations with different decay lengths on the same original images, which would make it easier to compare.

---

> ### Author Response · Authors · 2018-11-22
> **Detailed reply, highlighting our main contributions (part 1)**
>
> We would like to thank the reviewer for his/her valuable feedback.
>
> - PGD attack iterations -
> Regarding PGD iterations, we would like to quote [Madry A. et al. Towards deep learning models resistant to adversarial attacks, 2017.], who reported whitebox attack accuracies for PGD with ** 7 iterations ** (see Table 2): “For the CIFAR10 dataset, [...] we trained the network against a PGD adversary with l_infty projected gradient descent again, this time using 7 steps of size 2, and a total ε = 8.” That said, we don’t think that more iterations and random restarts would change the qualitative picture of our evaluations.
>
> - Attack accuracy vs area under the attack curve -
> Reporting area under the attack curve serves two purposes: Firstly, it addresses the potential danger of overfitting to a specific epsilon attack. Secondly, it mimics the realistic scenario in which the attacker tries to fool the classifier with as small a perturbation as possible. That said, we believe that an even more realistic performance measure would give less weight to larger perturbations that are easier to detect and give relatively more weight to smaller ones that are harder to detect.
>
> - Robustness under the strongest whitebox attack should be the benchmark -
> We disagree with this statement for two reasons. Firstly, without reference to an attack “budget”, more precisely a (distributional) uncertainty set as well as an upper bound on computational resources to search for worst case perturbations, the notion of “strongest” is ill-defined. Even if we agree on a computational budget, the question remains of how to define or measure the strength of perturbations - norm-based, perceptually similar, etc. Secondly, robustness comes at a price: rather than aiming for robustness against the strongest attack, we believe that one should aim for an optimal trade-off between robustness and clean accuracy. In that sense, it is debatable whether training methods that considerably reduce clean accuracy even deserve to be called robust. It is worth noting that this latter point has long been understood in the statistics community, see for instance P.J. Huber’s book on Robust Statistics.
>
> - Adversarial robustness via integrating over perturbations -
> We propose to efficiently approximate expectations over corrupted distributions through structure-informed regularization, as outlined in Section 2.2 (see Equation 3) and Section 3. Conceptually, our starting point is to learn a corruption model, i.e. to use a generative mechanism to learn adversarial perturbations from examples. ** Integrating over these corruptions is not the same as integrating over the neighborhood, however **. The intuition is that if the model is robust against the entire distribution of perturbations, it should also be robust against point-wise perturbations (from which the corruption model was learned). In practice, we propose to approximate such a corruption model by adaptively learning the structure of adversarial perturbations.
>
> One of the main contributions of our work is to ** derive structured gradient regularization ** as a tractable approximation to training with correlated perturbations. This is in line with a large body of work on the equivalence between regularization and robust optimization. See our reply titled "Our work is a strict generalization of previous work and regularization was proven to be equivalent to robust optimization in certain settings." for a list of references.
>
> - Bayes optimal classifier -
> The underlying assumption is that Eq. (10) and Eq. (5) coincide to order O(||\xi ||^3) at the optimum, which is within the precision to which we truncate. This assumption is rather common in the literature, see e.g. [Bishop. Training with noise is equivalent to tikhonov regularization., 1995] or [An, G. The Effects of Adding Noise During Backpropagation Training on a Generalization Performance. 1996]. Alternatively, Eq. (10) can also be seen as a Levenberg-Marquart approximation of Eq. (5), if one does not want to invoke the Bayes optimality argument, see Section 5.4.1 in Bishop’s Pattern Recognition and Machine Learning book.
>
> - Covariance structure too coarse as a measure of attack power? -
> Whether or not covariance structure is a good measure to distinguish different attacks depends on the entirety of attacks under consideration. It could be that both PGD and FGSM are members of the same “universality class” of adversarial attacks. After all, if we compare those two attacks with the entirety of all imaginable attacks, they are probably rather similar compared to other, e.g. gradient-free attacks. Nevertheless, we agree that it would be interesting to further explore the connection between covariance structure and attack power.

---

> > ### Author Response · Authors · 2018-11-22
> > **Detailed reply, highlighting our main contributions (part 2)**
> >
> > - Centered vs. uncentered corruption model -
> > Indeed, centered vs. uncentered distribution of perturbations refers to whether E_Q[\xi] is zero or not, as stated in Equation 12. We empirically observed that the mean adversarial perturbation is very close to zero, which is why we used the centered SGR regularizer in Equation 11 in all our experiments.
> >
> > - Figure 5: Long-range structured covariance matrices for increasing decay lengths -
> > The covariance matrices in Figure 5 were generated according to the intra-channel and inter-channel covariance functions discussed in Section 4.3. The periodic patterns are in part a result of the fact that the 2D image is first flattened into a 1D vector in order to plot the covariance matrix. Visual patterns then emerge because correlations for pixels at opposite ends of two neighboring rows are plotted next to each other due to the flattening..
> >
> > - Long-range correlated attack -
> > The LRC attack is indeed a sampling-based natural prototype for low frequency perturbations. See also our reply titled "The purpose of the LRC attack experiment is to establish whether there is a potential benefit in using a structured covariance matrix in the SGR regularizer."

---

### Official Review · AnonReviewer2 · 2018-11-01
**Some interesting ideas but unconvincing empirical evaluation**

**Rating:** 4
**Confidence:** 4

**Review:**

Short paper summary: This work proposes a novel method of gradient regularization (SGR) which utilizes the covariance structure of adversarial examples generated during training. The authors propose simple techniques to reduce the computational overhead of SGR. Empirically, the authors compare their method to standard adversarial training and gradient norm regularization.

Brief review summary: There are some interesting ideas in this work but I feel that the some practical aspects lack formal justification and the comparison to existing work is inconclusive.

Detailed comments:

In addition to some minor comments, I have two concerns. First, with the SGR algorithm itself. And second with the empirical analysis. While I suspect that the first concern may be clarified with discussion I think that the second is more serious and is the primary factor behind my review score.

1) As the SGR algorithm is written I wonder whether the regularization term may be computed more efficiently using something like a Hutchinson trace estimation trick. I suspect that if the random vector used to estimate the trace was the xi from Algorithm 1 then the same Mahalanobis gradient norm would be recovered. This would hold only in the case beta=1, bringing me to my second point.

2) What is the purpose of the running average of the covariance? A relatively small beta value is used in practice but I do not see any strong justification for this. Is there a good reason why we do not want the covariance matrix to be a close approximation for the local gradient landscape? This seems like an important part of the algorithm, especially as it may shed light on my next note.

3) In practice, Algorithm 1 uses adversarial attack schemes to generate the perturbations. In simple cases like FGM, this would give the covariance of the input-output gradient which seems that it would have a direct interpretation as a form of classical gradient regularization. To this extent, I also wonder how the SGR algorithm could be related to interpretations of adversarial training as gradient smoothing (when using small perturbations).

I recognize that the above points are (so far as I could tell) not directly addressed in the work, and some may be fairly considered out of scope. However, due to the direct comparison to adversarial training later and the need to tie SGR to adversarial attacks I feel that it would be important to distinguish these cases.

Overall, I felt that the first three sections did well to introduce the motivation and techniques used and were was easy to follow. The derivation of the SGR algorithm was clear and concise but I believe that some of the practical details (covariance running average, computational efficiency [at first glance, it looks like the full Jacobian must be computed, but practically the sum over K reduces this to a single backprop call]) could have been elaborated on.

For the empirical evaluation the authors provided ample detail on the experimental set up and have performed a fairly thorough investigation in terms of existing defenses and attacks. I felt that the bulk of the study which is contained in Table 1 is fairly inconclusive or at the very least, difficult to interpret completely. Additional comments:

4) I felt that Figure 1 and 2 are a little difficult to interpret at first. It would help to clearly define what is meant by short- and long-range signal corruptions. However, they do suggest some interesting findings. As these covariance matrices depend directly on the model itself, I think it is worth investigate (or commenting on) how this structure may change when introducing things like SGR (or GN). The authors claim that unregularized classifiers give too much weight to short range correlations but they should show that SGN (or other methods) correct this.

5) My biggest concern with this work is with the results presented in Table 1. In terms of how they are presented: first I think that the fool column requires further explanation, or perhaps more simply the column could show accuracy instead of the average perturbation size. Second, I am not sure why the reported accuracies are averaged over attack strengths in a range. So far as I am aware, this is not standard and makes it difficult to interpret the performance of the models in this way. Figure 4 in the appendix does a better job of describing the behavior over a range of attack strengths.

6) From the table, it is not obvious to me that SGR provides any improvements to robustness over existing techniques. Indeed, the authors write that SGR achieves white-box accuracies which are between those of the clean and adversarially trained models and claim that SGR improves on the clean accuracy for CIFAR-10. But in the table the gap between FGSM and GN/SGR clean accuracies seem fairly small with FGSM providing better robustness (for most source attacks). Even more concerning, is the fact that GN seems to outperform SGR. I do not find these results substantial enough to motivate SGR as a robustness defense compared with adversarial training (or even GN), especially as SGR has the same computational limitations involved with expensive adversarial perturbations.


I felt that the study into the covariance structure of adversarial perturbations was interesting but as it stands was not complete enough to be informative in general. In the conclusion the authors write that they provide evidence that current adversarial attacks act by perturbing the short-range correlations of signals but this has only been confirmed for unregularized classifiers. Despite these issues, I thought that the paper was well written and hope that the empirical study can be improved and clarified.


Minor comments:

- Section 2.1, set of transformations only introduced briefly then forgotten. Leaving output invariant confused me, as this does not apply to adversarial examples.
- Section 2.3, second paragraph l3: In Maaten et al. should be citet.
- Section 3.1, should  make clear that derivative is with respect to the data.
- Section 3.1, define delta as the Hessian clearly (it is used for the simplex in the previous section). Though this is easy to figure out.
- Section 7.1, starts with (iii), is this intentional? Perhaps an introductory sentence could make this clearer.
- Section 7.3, for label leaking, I'm not convinced by this argument alone. Assuming the covariance structure is still computed from a particular adversarial example, I see no compelling reason that this would not occur.


Clarity: The paper is very clearly written and is easy to follow.

---

> ### Author Response · Authors · 2018-11-22
> **Detailed reply, highlighting our main contributions (part 1)**
>
> We would like to thank the reviewer for his/her valuable feedback.
>
> 1) Hutchinson trace estimation trick
> The Hutchinson trace estimation trick doesn’t seem to be relevant for our regularizer: we are not primarily concerned with the problem of estimating the trace of the covariance matrix, but we are rather interested in leveraging the sparseness of the covariance-gradient matrix-vector product. Irrespective of that, we can already efficiently aggregate batch estimates for the covariance structure in our regularizer, as the input gradient of the per-sample cross-entropy loss is often available as a highly optimized callable operation in modern deep learning frameworks. Nevertheless, it is an interesting suggestion which we would be happy to investigate further.
>
> 2) What is the purpose of the running average in the covariance?
> The decay rate β allows us to trade off weighting between current (β → 1) and past (β → 0) batch averages. The idea of using smaller decay rates is that this should avoid overfitting to a specific attack: the more of the history we take into account (i.e. the more momentum), the less likely the model is to overfit on specific perturbations. Our choice of β=0.1 was inspired by momentum-based adaptive optimization algorithms like Adam, which also by default gives a weight of 0.1 to current gradients and a weight of 0.9 to past gradients. We did not observe a big difference in our experiments for other values of β.
>
> 3) SGR algorithm vs. adversarial training as gradient smoothing
> Our regularizer is informed by the covariance structure of adversarial perturbations, which for simple perturbations, like FGM, is indeed given by the covariance of the input-output gradient. That said, it seems well worth exploring whether adversarial training can be interpreted as gradient smoothing and how this is connected to SGR regularization.
>
> 4) Covariance structure of adversarial perturbations and how it might change
> The decay length is defined as the displacement over which the covariance function decays to 1/e of its value. The covariance function is just a simple parametrization of the covariance matrix in terms of the displacement between pixels, as is well-known in computer vision. Based on the observation that unregularized/undefended classifiers are vulnerable to short-range structured corruptions, we thus conjecture that they give too much weight to short-range correlations (high-frequency patterns) and not enough weight to long-range ones (globally relevant low-frequency features).
>
> The question of how this structure may change when robustifying the model through adversarial training or SGR regularization is indeed interesting. What makes this analysis complicated, however, is the fact that the ** covariance structure not only depends on the model but also on the attack algorithm **. So, if the model becomes more robust to short-range correlated perturbations, the following two things can happen (potentially both): (i) new perturbations become less effective and thus more random, in which case the decay-length of the covariance function becomes even shorter. Or (ii) the attack will adapt to perturb the long-range (low-frequency) content of the signal, if it is powerful enough. Assessing the covariance function change therefore seems rather non-trivial, as one would need to separate the effect of model robustness from attack algorithm adaptivity/non-adaptivity. We did not observe meaningful changes of the covariance structure in our experiments, which is not a negative result due to the above points however.
>
> 5) White-box and transfer attack accuracy results
> The DeepFool attack is unconstrained: if it is run for sufficiently many iterations, it should always reduce the accuracy of the classifier to below chance. This is why the Fool column in Table 1 reports the magnitudes of the perturbations required to cross the decision boundary (normalized by the magnitude of the unperturbed data point), according to Equation 2 (or its empirical counterpart in Equation 15) in [Moosavi-Dezfooli et al, DeepFool: A Simple and Accurate Method to Fool Deep Neural Networks, 2016].
>
> Reporting area under the attack curve serves two purposes. Firstly, it addresses the potential danger of overfitting to a specific attack epsilon. Secondly, it mimics the realistic scenario in which the attacker tries to fool the classifier with as small a perturbation as possible. That said, we believe that an even more realistic performance measure would give less weight to larger perturbations that are easier to detect and give relatively more weight to smaller ones that are harder to detect. (Note, the numbers we report give equal weight to different perturbation strengths.)

---

> > ### Author Response · Authors · 2018-11-22
> > **Detailed reply, highlighting our main contributions (part 2)**
> >
> > 6) White-box and transfer attack accuracy results (II)
> > As stated in Section 4.4, SGR/GN trained models achieve white-box attack accuracies that are intermediate between those of the clean model and adversarially trained models. We would like to note, however, that we do not equate “robustness” with “white-box attack accuracy”. If we look at the transfer-attack accuracies (bold-face numbers), then SGR and GN trained models are statistically on par with adversarially trained models.
> >
> > We would also like to add that the PGD white-box attack accuracies reported for SGR and GN trained models are within one standard deviation of each other, which is 0.5 % (computed over 10 runs). We can therefore only conclude that SGR and GN trained models achieve statistically indistinguishable accuracies.
> >
> > Minor comments:
> > - The purpose of data augmentation is to induce invariance of the output (i.e. the classifier predictions) w.r.t. a set of input transformations. A robust classifier should - to some extent - also be invariant to adversarial examples.
> > - Section 7.1 should start without the (iii) typo.
> > - Even if the covariance structure is computed from one single example, the SGR regularized classifier is only ever evaluated on the clean input, i.e. adversarial perturbations are never fed to the classifier. It thus seems impossible that the classifier performs better on perturbed examples than on clean inputs and in practice we also did not observe this.

---

> > > ### Comment · AnonReviewer2 · 2018-11-26
> > > **Thank you for clarifications I**
> > >
> > > Thank you for your detailed response and my apologies for writing my own later than should be acceptable.
> > >
> > > 1) I don't see this as an especially pivotal part of my review. My point was that computing the quadratic form required for the SGR regularizer may be more computationally efficient if you take advantage of the structure of the covariance matrix - avoiding computing it directly. When using the running average I don't see how this could be easily achieved.
> > >
> > > 2) Thank you for clarifying. Unfortunately, this still seems like quite a weak argument to me (though intuitively it makes sense). Is there anything from the regularization literature that you cite which may help to justify this technique preventing overfitting? Is this something that you could justify empirically? I know that we are past the revision date at this point so I want ensure you that this is not a critical part of my review but is something I would be interested to hear your thoughts on.
> > >
> > > 3) I think this is an important point and one that does require further thinking. It seems to me that in some cases the SGR algorithm will reduce to a small-epsilon form of adversarial training. In this case - what is special about SGR that has it outperform adversarial training? See [1,2] for some description of how adversarial training can be intepreted as gradient smoothing. This is fairly easy to see by looking at f(x+d) - f(x), for d given by e.g. single gradient steps in the limit of small perturbations.
> > >
> > > 4) Thank you for clarifying. I still don't see this explanation in the main paper (or an accompanying citation). Am I missing this? I think it would be reasonable to include some description for those of us who aren't overly familiar with this terminology.
> > >
> > > I still want to put emphasis behind a comment from my initial review. You hypothesise that putting too much emphasis on short-range correlations leads to vulnerability. But you do not test the more interesting converse at all: reducing dependence on short-range correlations improves robustness. Nor do you show that SGR is able to reduce dependence on short-range correlations. You raise two points here (denoted i and ii) which seem interesting and important to me. If (i) holds, then does this indicate that a stronger attack may work even better against the model? If not, then why is decay-length still a meaningful indicator of robustness? I think that (ii) seems potentially more interesting, perturbing the low-frequency features (if I understand correctly) would have some effect on the semantic meaning of the perturbation - similar to that observed in adversarial training. [3]
> > >
> > > To me, this is an important part of the theoretical discussion in this paper but it is underexplored - both empirically and analytically. I acknowledge that there may be difficulty when disentangling the covariance structure due to the model and attack but if this is the case then it seems unreasonable to conclude that dependence on short-range correlations => vulnerability.

---

> > > > ### Comment · AnonReviewer2 · 2018-11-26
> > > > **Thank you for clarifications II**
> > > >
> > > > 5) Thank you for clarifying the DeepFool column. Though I don't have any immediate suggestions, it seems that this has been a source of confusion to other readers and should probably be addressed. Maybe further explanation in the text?
> > > >
> > > > I largely agree with your comments on AUC - perhaps this would be a better measure. However, I still believe that this makes comparing to existing work more difficult. Perhaps Figure 4 could be produced for a few models and pointed to in the main text (so that the new figures remain in the appendix, if you prefer).
> > > >
> > > > I don't see statistical parity with adversarial training as especially exciting - especially as robust optimization adversarial training is not included [4]. My biggest concern with the work still lies in the soundness of the empirical study. I do not feel that sufficient evidence has been provided to recommend decay length of signal corruptions as a good measure of robustness (or attack strength) but there are some interesting findings here that I would like to see explored further. I am also unconvinced by the results presented for SGR, in particular that it does not seem to offer any advantage over GN regularization.
> > > >
> > > > Response to minor comment:
> > > >
> > > > > Even if the covariance structure is computed from one single example, the SGR regularized classifier is only ever evaluated on the clean input, i.e. adversarial perturbations are never fed to the classifier. It thus seems impossible that the classifier performs better on perturbed examples than on clean inputs and in practice we also did not observe this.
> > > >
> > > > This seems like a subtle point. The classifier is used to produce the adversarial perturbations which build the covariance matrix. The computation graph is then "broken" so that no gradient is passed through the network using these perturbations, but the covariance matrix is used as a regularization term. From comment (3) above it feels that in some special cases this may end up looking very similar to existing approaches that use gradient smoothing/adversarial training (minus the covariance running average). In summary, it still isn't obvious to me that overfitting is impossible. If you only learn the covariance structure of single step gradient attacks local to each traning datapoint how can you argue generalization to new attacks (higher order, new threat models e.g. L2 vs L infinity, decision based attacks, transfer attacks) on test data?
> > > >
> > > >
> > > > Short summary: There are some interesting parts to this work but I feel that there is insufficient evidence to support these. My issue still lies mostly with the empirical evaluation.
> > > >
> > > > [1] Simon-Gabriel et al. "Adversarial vulnerability of neural networks increases with input dimension" https://arxiv.org/pdf/1802.01421.pdf
> > > > [2] Miyato et al. "Virtual adversarial training: A regularization method for supervised and semi-supervised learning" https://arxiv.org/abs/1704.03976
> > > > [3] Tsipras et al. "There Is No Free Lunch In Adversarial Robustness (But There Are Unexpected Benefits" https://arxiv.org/abs/1805.12152v2
> > > > [4] Madry et al. "Towards deep learning models resistant to adversarial attacks" https://arxiv.org/abs/1706.06083

---

### Official Review · AnonReviewer3 · 2018-11-03
**Trying to address an important problem, but approach/results are not convincing**

**Rating:** 4
**Confidence:** 4

**Review:**

The authors propose a new defense against adversarial examples that relies on a data-dependent regularization (instead of adversarial training). They then benchmark the performance of this new defense against popular white-box and transfer attacks, as well as propose a new long range correlated adversarial attack.

Comments:
I find the premise of this paper interesting - developing regularization strategies to help with generalization to adversarial perturbations. For instance, it is well known that state-of-the-art defenses such as PGD have generalization gaps as large as 50% between robust train and test accuracies. It has also been previously hypothesized that this could be due to a data scarcity problem [Schmidt et al., 2018].  The authors here propose to tackle this problem using a new data-dependent regularization technique.

My primary issue with this paper is that the authors do not clearly illustrate what the advantage of their method over standard methods is
- The problem this paper aims to solve is overfitting to a specific attack/virtual adversarial examples presented during adversarial training by using regularization instead. However, the authors do not actually illustrate that their technique reduces overfitting. For instance, the authors do not contrast the robust train-test accuracies using their method to other standard methods. Thus it is not clear that this paper met the objectives laid out in the introduction.
- The claim in this paper is that SGR helps against attacks with long range dependencies. However, in their experiments (e.g., in Figure 3), the authors do not evaluate other standard defenses. It is thus unclear whether other standard methods are already robust to such attacks. In fact, based on the results of Table 1, it doesn’t seem like attacks from SGR  are able to reduce the robustness of PGD/FGSM trained models.

Because of these two points, along with the lower robustness to various attacks (in Table 1) as compared to approaches such as PGD, it is not really clear to me what the real merit of this new approach is. Ultimately, having a defense which is more robust to a particular attack is not very meaningful if there exists an alternative attack that reduces the robustness of the defense.

I am also surprised that the authors chose to use this regularization as an alternative to adversarial training instead of complementary to it. I would be interested to see if such regularization could actually help to bridge the generalization gap observed while using adversarial training.

The paper is at times is poorly written and confusing. For instance, the description of CovFun is hard to parse. The authors should make this explanation more clear. The authors also do not state what their attack model is - Linf vs L2 perturbations. They also choose to evaluate attacks differently, using an average accuracy over different epsilons rather than reporting individual accuracies. This does make the results harder to compare to other work. The authors should include a full table of individual accuracies (at least in the appendix) to make the numbers easier to parse and compare.

In the derivation in Section 3.1, the authors use the assumption that the robust classifier is almost equal to the Bayes optimal classifier to justify dropping terms corresponding to the Hessian(\phi_y). I am not sure how realistic this assumption is in the adversarial setting - one can construct simple distributions for which the Bayes optimal classifier is not the robust classifier.

With regards to Figure 3, the authors state -
“As the decay length goes to zero, the synthetic covariance matrix converges to the identity matrix and SGR performance approaches GN performance”
Could the authors clarify why this is obvious? After all these two models are trained very differently.

The plot in Figure 3 and the results in Table 1 seems to illustrate that SGR is no better than GN as you can find an attack where they perform as well/badly. The authors say that this is due to the short-range nature of current attacks. I do not understand this rationale though - the goal of the defenses should be to be more robust to all attacks, both short range and long range. Thus arguing that there may be an attack under which their model performs better is not sufficient. I do agree that finding long range attacks that can break current SOTA robust models would be interesting, however the authors do not seem to achieve that in this work.

I find the observation on transfer attacks interesting - PGD attacks from SGR/GN models are better than PGD models. Do the authors have any insight as to why this is the case?

In general, my concern about gradient regularization based defenses is that they only give a very local picture of the landscape and thus can only protect against small eps attacks. This could probably explain why the SGR/GN models are less robust than PGD. As mentioned previously, it would be valuable to see accuracies against individual eps values (rather than averaged) to understand this better. If this is the case, this regularization would not provide any additional benefits when combined with adversarial training either.

References:
Schmidt, Ludwig, et al. "Adversarially Robust Generalization Requires More Data." arXiv preprint arXiv:1804.11285 (2018).

---

> ### Author Response · Authors · 2018-11-22
> **Detailed reply, highlighting our main contributions (part 1)**
>
> We would like to thank the reviewer for his/her valuable feedback.
>
> - Merits of structured gradient regularization -
> SGR has several conceptual merits:
>
> Firstly, one of the main contributions of our work is to ** derive structured gradient regularization ** as a tractable approximation to training with correlated perturbations. SGR is a generalization of gradient norm (GN) regularization: while GN provides an approximation to training with white noise, SGR provides an approximation to training with arbitrarily correlated noise. This is in line with a large body of work on the equivalence between regularization and robust optimization. See our reply titled "Our work is a strict generalization of previous work and regularization was proven to be equivalent to robust optimization in certain settings." for a list of references.
>
> Secondly, while robust optimization aims at approximating the worst-case distribution, we propose to efficiently approximate expectations over corrupted distributions through structure-informed regularization. Conceptually, rather than perturbing each data point individually, our starting point is to learn a corruption model, i.e. to use a generative mechanism to learn adversarial perturbations from examples. In practice, we propose to approximate such a corruption model by adaptively learning the structure of adversarial perturbations.
>
> Thirdly, SGR can leverage the fact that adversarial examples might live in low-dimensional subspaces. Quoting from [Moosavi-Dezfooli et al, “Universal adversarial perturbations”, 2017]: “We hypothesize that the existence of universal perturbations fooling most natural images is partly due to the existence of such a low-dimensional subspace that captures the correlations among different regions of the decision boundary.” SGR can leverage this by penalizing gradients that lie within such a subspace.
>
> - Combining SGR with adversarial training -
> It has certainly occurred to us to combine SGR with adversarial training. However, in the interest of transparency, we believe it is more clear to benchmark and compare regularization and adversarial training individually. Nevertheless, we will investigate combining them.
>
> - Covariance function -
> The covariance function is just a simple parametrization of the covariance matrix in terms of the displacement between pixels, as is well-known in computer vision. We apologize for omitting to specify that the PGD attack was L_infty constrained.
>
> - Attack accuracy vs area under the attack curve -
> Reporting area under the attack curve serves two purposes. Firstly, it addresses the potential danger of overfitting to a specific attack epsilon. Secondly, it mimics the realistic scenario in which the attacker tries to fool the classifier with as small a perturbation as possible. That said, we believe that an even more realistic performance measure would give less weight to larger perturbations that are easier to detect and give relatively more weight to smaller ones that are harder to detect. (Note, the numbers we currently report give equal weight to different perturbation strengths.)
>
> - Cancellation of Laplacian terms -
> The underlying assumption is that Eq. (10) and Eq. (5) coincide to order O(||\xi ||^3) at the Bayes optimum, which is within the precision to which we truncate. This assumption is rather common in the literature, see e.g. [Bishop. Training with noise is equivalent to tikhonov regularization., 1995] or [An, G. The Effects of Adding Noise During Backpropagation Training on a Generalization Performance. 1996]. Alternatively, Eq. (10) can also be seen as a Levenberg-Marquart approximation of Eq. (5), if one does not want to invoke the Bayes optimality argument, see Section 5.4.1 in Bishop’s Pattern Recognition and Machine Learning book.
>
> - Long-range correlated noise attack -
> We do not claim that the LRC attack can break existing methods. The purpose of the LRC attack experiment is solely to establish whether there is a potential benefit in using a structured covariance matrix in the SGR regularizer versus using an “unstructured” diagonal covariance (corresponding to gradient-norm regularization) in the presence of long-range correlated noise. In other words, this experiment simply tests whether the SGR regularizer extracts useful information about the long-range correlation structure of the perturbations, which it indeed does.
>
> - Decay length approaching zero -
> The quoted statement is indeed trivial: if SGR is trained from scratch with a covariance matrix that is close to the identity matrix (i.e. the covariance matrix has a decay length close to zero), its performance will be similar to that of GN, as shown in Figure 3. Note, that each data point in Figure 3 corresponds to (an average of five) networks that have been trained from scratch with a covariance matrix of the given decay length.

---

> > ### Author Response · Authors · 2018-11-22
> > **Detailed reply, highlighting our main contributions (part 2)**
> >
> > - SGR/GN white-box and transfer attack accuracies -
> > As stated in Section 4.4, SGR/GN trained models achieve white-box attack accuracies that are intermediate between those of the clean model and adversarially trained models. We would like to note, however, that we do not equate “robustness” with “white-box attack accuracy”. If we look at the transfer-attack accuracies (bold-face numbers), then SGR and GN trained models are statistically on par with adversarially trained models.
> >
> > We would also like to add that the PGD white-box attack accuracies reported for SGR and GN trained models are within one standard deviation of each other, which is 0.5 % (computed over 10 runs). We can therefore only conclude that SGR and GN trained models achieve statistically indistinguishable accuracies.
> >
> > - Transfer attack strength -
> > This could in part be due to PGD adversarial training resulting in adversarial perturbations that become easier to classify instead of the classifier actually becoming more robust. See for instance [Athalye et al. Obfuscated gradients give a false sense of security, 2018.] or [Galloway et al., Adversarial training versus weight decay, 2018].
> >
> > - Are gradient regularization based defenses only giving a very local picture of the landscape? -
> > Not necessarily. If adversarial vulnerability is an intrinsic property of the network, regularization as well as other adversarial robustification methods might remedy this vulnerability without having to search for adversarial perturbations in a certain neighborhood around each data point in the first place.
> >
> > - Evidence that SGR reduces overfitting -
> > We did compute numbers for the training accuracy - test accuracy generalization gap for the various training methods considered in our paper. What we see is that clean, PGD and FGSM trained models have generalization gaps of around 10-14% whereas GN and SGR trained models have generalization gaps of around 5-7%.

---

> > > ### Comment · AnonReviewer3 · 2018-11-27
> > > **Thank you for the clarifications**
> > >
> > > I thank the authors for their clarifications and apologize for my delayed response.
> > >
> > > - Merits of structured gradient regularization -
> > > I agree that SGR has some potential conceptual merits, but the studies in this paper are not yet sufficient to demonstrate that these merits translate into practice. More broadly, I believe that regularization approaches for robust learning could indeed have many benefits, in terms of - (1) improving the generalization performance (2) offering a computationally less expensive alternative for adversarial training (by not requiring adversarial examples to be computed by an involved process like PGD) or (3) lower the sample complexity required in robust learning. However, the paper, in its current version does not provide convincing evidence on any of these fronts.
> > >
> > > - Combining SGR with adversarial training -
> > > I think this investigation is important to establish the merits of this approach, in the light of the other empirical results. In particular, I believe it would be really valuable if the generalization gap observed between train and test adversarial accuracies with adversarial training is decreased when you train with adversarial training + SGR.
> > >
> > > - Covariance function -
> > > I thank the authors for this clarification.
> > >
> > > - Attack accuracy vs area under the attack curve -
> > > I agree that reporting AUC may have merits as an evaluation approach. However, as this is not standard in the robustness literature, I think it is essential for the authors to also include the results without averaging to make it easier to evaluate in the light of prior work.
> > >
> > > - Cancellation of Laplacian terms -
> > > I thank the authors for the clarification. But I do not agree that these properties that hold for training with *white* noise or in the standard setting can be claimed (without further analysis) to hold in the adversarial setting.
> > >
> > > - Long-range correlated noise attack -
> > > I think the idea of investigating the structure of attacks proposed in this paper is interesting. But it warrants further exploration. For instance, I would like to see how state-of-the-art robust models do wrt these LRC attacks. I also agree with AnonReviewer2’s comments that an investigation on the relationship between the structure of attacks and robustness warrants a deeper theoretical and empirical investigation.
> > >
> > > - Decay length approaching zero -
> > > Thank you for clarifying.
> > >
> > > - SGR/GN white-box and transfer attack accuracies -
> > > I thank the authors for the clarification, but I am still not convinced by these results. As I mentioned in my response to “- Merits of structured gradient regularization -” above, I think there are multiple avenues to demonstrate the merits of SGR as a defense (if it does match SOTA approaches currently), but I do not think they have been sufficiently demonstrated in this paper.
> > >
> > > - Evidence that SGR reduces overfitting -
> > > Could the authors include these results in the manuscript?
> > >
> > > I think this paper tackles an important question and raises some interesting points (about the relationship between the structure of adversarial perturbations and robustness). However these have not been sufficiently explored in the paper and I find the empirical investigation lacking.

---

### Public Comment · (anonymous) · 2018-10-03
**How is this significantly different from previous broken defenses based on gradient regularization?**

Regularizing the norm of the gradient of the output log probability with respect to the input has been tried many times and does not work as a defense against adversarial examples.

This work essentially proposes to use a Mahalanobis norm ( g^T A g) rather than a squared L2 norm (g^T g) for the gradient penalty. Why would this be any better?

Gradient regularization does not work because it is based on derivatives and thus is designed to resist only infinitesimal perturbations. It cannot "see" the way that finite-sized perturbations cross relu boundaries and so on. Using a Mahalanobis norm rather than an L2 norm doesn't address this fundamental limitation of gradient regularization. All it does is penalize the gradient more in some directions than others.

If anything, using a Mahalanobis norm seems like it should create more opportunities for adversarial attacks to succeed in the directions that were downweighted.

---

> ### Author Response · Authors · 2018-10-06
> **Our work is a strict generalization of previous work and regularization was proven to be equivalent to robust optimization in certain settings.**
>
> - How is this significantly different from previous defenses based on gradient regularization?
>
> To the best of our knowledge, gradient regularization as a method to improve adversarial robustness has been studied in two other concurrent works, both of them are cited [A.S. Ross & F. Doshi-Velez, “Improving Adversarial Robustness and Interpretability of DNNs by Regularizing Input Gradients”, 2017] and [C.J. Simon-Gabriel et al. “Adversarial Vulnerability of Neural Networks Increases With Input Dimension”, 2018].
>
> Firstly, our work goes a lot further in terms of theoretical justification for gradient regularization than both of these: we follow a principled approach to derive structured gradient regularization as a tractable approximation to training with correlated perturbations.
>
> Secondly, our structured gradient regularizer (SGR) is a strict generalization of gradient norm (GN) regularization: while GN provides an approximation to training with white noise, SGR provides an approximation to training with arbitrarily correlated noise. Moreover, regularization has been shown to be equivalent to robust optimization in certain settings, see below.
>
>
> - Why would gradient regularization w.r.t. Mahalanobis distance (i.e. SGR) be any better than gradient regularization w.r.t. L2 norm?
>
> Firstly, gradient norm regularization based on L2 norm assumes isotropic white-noise, whereas Mahalanobis-distance based SGR operates with arbitrarily correlated noise.
>
> Secondly, SGR can leverage the fact that adversarial examples might live in low-dimensional subspaces. Quoting from [Moosavi-Dezfooli et al, “Universal adversarial perturbations”, 2017]: “We hypothesize that the existence of universal perturbations fooling most natural images is partly due to the existence of such a low-dimensional sub-space that captures the correlations among different regions of the decision boundary.” SGR can leverage this by penalizing gradients that lie within such a subspace more strongly than gradients that lie outside it.
>
>
> - Gradient-norm regularization has been tried many times and does not work as a defense against adversarial examples.
>
> First of all, could you please provide references to papers where gradient regularization was the main method of defense (i.e. where it was not just used as a baseline) and was shown not to work?
>
> Secondly, this statement is unqualified: to be precise, you need to (i) state how you measure performance, i.e. how you define whether some method “works” and (ii) what kind of threat model you assume, i.e. what kind of “adversarial examples” you want to robustify against. E.g. gradient-based or gradient-free, white-box or transfer/black-box attacks, whether the perturbations are constrained in magnitude or whether they are constrained by the counting-norm etc. In your statement you seem to make specific assumptions, which is why it is not true in the generality in which it was formulated.
>
> For instance, if we take transfer attack accuracies as our measure of robustness and PGD transfer attacks as the threat-model, corresponding to the bold-face numbers in Table 1, then SGR and GN are statistically on par with PGD and FGSM trained models on CIFAR10, compare the bold-face numbers in each row.
>
>
> - Gradient regularization does not work because it is based on derivatives and thus is designed to resist only infinitesimal perturbations.
>
> There is a large body of work on the equivalence of regularization and robust optimization (adversarial training is a special case of robust optimization against a pointwise adversary that independently perturbs each example):
>
> [Bertsimas and Copenhaver, “Characterization of the equivalence of robustification and regularization in linear and matrix regression” 2018], showed that in linear regression robust optimization for matrix-norm uncertainty sets and regularization are exactly equivalent. There is also a variety of settings for robust optimization under more general uncertainty sets in which regularization provides upper and lower bounds. See also [El Ghaoui and Lebret, “Robust solutions to least-squares problems with uncertain data” 1997].
>
> [Xu et al., “Robustness and regularization of support vector machines“ 2009] established equivalence of robust optimization and regularization for Support Vector Machines.
>
> More recently, [Gao et al., “Wasserstein distributional robustness and regularization in statistical learning” 2017] showed that Wasserstein-distance based distributionally robust stochastic optimization (Wasserstein-DRSO) is first order equivalent to gradient regularization.
>
> These works clearly contradict your statement that gradient regularization does not work.

---

### Public Comment · (anonymous) · 2018-10-03
**Do you report attack success rate for a specific epsilon?**

Table 1 apparently shows areas under attack curves for varying epsilon ("The white-box and transfer attack accuracies are averaged over attack strengths in the range ∈ [0, 32] for MNIST and  ∈ [0, 8] for CIFAR10, i.e. the reported accuracies represent the integrated area under the attack curve."). This makes it hard to compare to previous work such as Madry et al 2017, who report attack success rate for the largest value of epsilon. Does the paper report the attack success rate for epsilon=8 specifically?

---

> ### Author Response · Authors · 2018-10-06
> **Please read the main text, we give a very precise explanation about what we report and we establish fair comparisons to the best of our abilities.**
>
> Firstly, we would like to emphasize that the text is very precise about what the numbers reported in the tables. As stated in the first paragraph of Section 4.4 as well as in the caption of Table 1, we report white-box and transfer attack accuracies averaged over attack strengths in the range [0, 32] for MNIST and [0, 8] for CIFAR10.
>
> Secondly, we establish a fair comparison between regularized models and adversarially trained ones, in that we train each architecture with various different training methods, including PGD-augmented training suggested in Madry et al. ***. In fact, for PGD and FGSM adversarial training, we trained models with each integer epsilon in the range [0, 32] for MNIST and [0, 8] for CIFAR10 and report results for the best performing one. The hyperparameters of the best performing models are reported in Section 7.2 in the Appendix.
>
> To the best of our knowledge, Madry et al. used different architectures and possibly different data preprocessing and data augmentation schemes.
>
> As a side note, we believe that an even more realistic performance measure should give less weight to larger perturbations which are easier to detect and give relatively more weight to smaller ones that are harder to detect. The averaged attack accuracies we report give equal weight to different perturbation strengths.
>
>
> ***: We assume that by “Madry et al. 2017” you meant [Madry et al., “Towards Deep Learning Models Resistant to Adversarial Attacks” 2017]

---

### Public Comment · (anonymous) · 2018-10-03
**The motivation for introducing the long-range correlated noise attack seems backward**

If I understand section 4.3 correctly, you introduce the LRC attack because you expect that your proposed defense will be able to beat it. This is not the way that you should evaluate new defense papers. New defenses should perform well against pre-existing attacks. Papers on new defenses sometimes need to introduce new attacks, but these should be new attacks that are *hard* for the defense to beat, not attacks that are designed to be *easy* for the defense to beat. For example, a new defense based on non-differentiable operations might perform poorly against pre-existing gradient-based attacks, so to evaluate it properly it is necessary to introduce new gradient-free attacks.

---

> ### Author Response · Authors · 2018-10-06
> **The purpose of the LRC attack experiment is to establish whether there is a potential benefit in using a structured covariance matrix in the SGR regularizer.**
>
> Thank you for raising this question for which we believe we need to reiterate several points.
>
> First of all, we did not design the LRC attack with the purpose to be easy to beat. If you read our submission carefully, you will notice that the purpose of the LRC attack experiment is to establish whether there is a potential benefit in using a structured covariance matrix in the SGR regularizer versus using an “unstructured” diagonal covariance (corresponding to gradient-norm regularization) in the presence of long-range correlated noise. In other words, this experiment simply tests whether the SGR regularizer extracts useful information about the long-range correlation structure of the perturbations, which it indeed does.
>
> Secondly, we do not claim that LRC is stronger than other pre-existing attacks. In your criticism, you seem to imply that we claimed that structured gradient regularization defends against pre-existing attacks because it performs well against long-range correlated perturbations, but we did not say that in our submission. Such a claim could be made if one showed that a new attack is stronger than existing ones and that a new defense protects against this new attack. We do not claim that however. Instead, and to the best of our ability, we transparently evaluate regularized and adversarially trained models against pre-existing white-box and transfer attacks in Section 4.4.
>
> The LRC attack is nothing but a natural prototype for low frequency perturbations, as opposed to existing attacks which we have shown to mainly corrupt the short range (high frequency) structure of signals. As stated in the conclusion, devising further (e.g. gradient-based) low frequency attacks is an interesting direction of future research.

---

### Public Comment · (anonymous) · 2018-10-03
**What is the "fool" column of table 1?**

The right way to read adversarial vulnerability tables is to take the min accuracy across different attacks: it doesn't matter if your defense is good at beating a lot of attack algorithms; if there is one attack that performs well then an attacker will use that.

In table 1 it looks like the "fool" attack is able to completely break the proposed defense, resulting in < 1% accuracy.

However, there are some other things that are weird. For example, the "fool" column also reports < 1% accuracy for a PGD-trained model. DeepFool is not previously known to break PGD-trained models, so this either indicates an interesting research finding, or a bug in your accuracy calculations, or a bug in your PGD-trained model.

---

> ### Author Response · Authors · 2018-10-06
> **The Fool column reports the noise-to-signal ratio of the DeepFool attack. Those numbers are not accuracies.**
>
> - The right way to read adversarial vulnerability tables is to take the min accuracy across different attacks
>
> We totally agree with this statement. We report various different attacks so that the reader can draw his or her own conclusions.
>
>
> - In table 1 it looks like the "fool" attack is able to completely break the proposed defense, resulting in < 1% accuracy.
>
>
> The numbers reported in the Fool column are not accuracies (which is why we did not use % sign but reported decimal numbers). If you read our paper carefully, the Fool column reports the noise-to-signal ratio of the DeepFool attack computed according to Eq.(2) in [Moosavi Dezfooli et al., “Deepfool: a simple and accurate method to fool deep neural networks.” 2016], as stated in the Experimental Setup Section 4.1.
>
>
> - We have checked our implementations and run many sanity-checks and we do believe they are correct. We do not see any evidence pointing to the contrary in our results.

---

### Public Comment · (anonymous) · 2018-10-03
**Table 1 shows the defense is worse than the baseline**

As mentioned in another comment, the "fool" column is strange, and if we can trust it, then all the defenses are shown to be completely broken.

If we ignore the fool column and just look at the other columns that seem more believable, then how does this model look on CIFAR-10?
The strongest attack against it is PGD, which results in an accuracy AUC of 41.5.
This is worse than the baseline of just doing gradient regularization with no Mahalanobis distance, which has a worst-case accuracy AUC of 41.9.
It is also worse than either of the two defenses based on adversarial training (which get 55 and 62 AUC).

We also see more or less the same thing on MNIST. Here the strongest attack against the proposed SGR defense is T-PGD, which gets 96.5 AUC. Traditional gradient regularization actually ties it, also with 96.5 AUC, just with a different attack causing the worst case performance. Both of the defenses based on adversarial training perform strictly better.

---

> ### Author Response · Authors · 2018-10-06
> **SGR and GN trained models achieve statistically indistinguishable attack accuracies (within one standard deviation of each other).**
>
> - The "fool" column is strange, and if we can trust it, then all the defenses are shown to be completely broken.
>
> See [Moosavi Dezfooli et al., “Deepfool: a simple and accurate method to fool deep neural networks.” 2016] on how to interpret those numbers correctly.
>
> - Worse than the baseline of just doing gradient regularization with no Mahalanobis distance?
>
> The PGD white-box attack accuracies reported for SGR and GN trained models are within one standard deviation of each other, which is $\sigma = 0.5$ (computed over 10 runs). What we can conclude from this table is that SGR and GN trained models achieve statistically indistinguishable accuracies for these particular results.
>
> As stated in Section 4.4, SGR/GN trained models achieve white-box attack accuracies that are intermediate between those of the clean model and adversarially trained models. Note, however, that we do not equate “robustness” with “white-box attack accuracy”. If we look at the transfer-attack accuracies (bold-face numbers), then SGR and GN trained models are statistically on par with adversarially trained models.

---

### Public Comment · ~Yifei_Wang1 · 2018-12-14
**Connection to natural gradient?**

I think the Structured Gradient Regularization you propose is very very similar to the classical **natural gradient** or the closely related **Gauss-Newton** method. In natural gradient they also approximate the deviation constraint by second order Taylor expansion (also drop higher order term in Hessian), resulting in the Fisher Information Matrix (FIM). The FIM term is then added back to the objective as a penalty, which is the same 'data-dependent' or 'structured' regularization in your paper. Indeed FIM could be seen as a local metric in the Riemann Manifold defined by current position so it's data-dependent. The only difference may lie in that, the natural gradient can have a closed form solution while you still utilize gradient descent (which solves the linearised objective), but this is no big deal.

---

### Meta-Review · Area_Chair1 · 2018-12-14

**Confidence:** 4
**Recommendation:** Reject

**Metareview:**

Reviewers are in a consensus and recommended to reject after engaging with the authors. Further, many additional questions raised in the discussion should be addressed in the submission to improve clarity. Please take reviewers' comments into consideration to improve your submission should you decide to resubmit.